# Sponge bHLH Gene Expression in *Xenopus laevis* Disrupts Inner Ear and Lateral Line Neurosensory Development and Otic Afferent Pathfinding

**DOI:** 10.3390/ijms26125487

**Published:** 2025-06-07

**Authors:** Karen L. Elliott, Clayton Gordy, Hannah Ingvalson, Charles Holliday, Jessica Halyko, Douglas W. Houston, Bernard M. Degnan, Bernd Fritzsch

**Affiliations:** 1Department of Biology, University of Iowa, Iowa City, IA 52245, USA; clayton-gordy@uiowa.edu (C.G.); hannah-ingvalson@uiowa.edu (H.I.); charles-holliday@uiowa.edu (C.H.); jessica-halyko@uiowa.edu (J.H.); douglas-houston@uiowa.edu (D.W.H.); 2School of Biological Sciences, University of Queensland, Brisbane 4072, Australia; b.degnan@uq.edu.au; 3Department of Neurological Sciences, University of Nebraska Medical Center, Omaha, NE 68198, USA

**Keywords:** *atonal*, bHLH, inner ear, lateral line, neurosensory development, *Neurod1*

## Abstract

Basic helix–loop–helix (bHLH) transcription factors, such as those in the *atonal* family, are important in cellular fate determination. The expression of the sponge ortholog of the *atonal* bHLH gene family, *AmqbHLH1*, in *Xenopus laevis* previously resulted in the formation of ectodermal ectopic neurons. However, the extent to which these neurons persist through development and the effects on the inner ear and lateral line, which require a critical level and timing of bHLH genes, remains unexplored. To test these long-term effects, we injected various concentrations of *AmqbHLH1* mRNA into *X. laevis* embryos and assessed neurosensory development at developmental stages coinciding with fully developed neurosensory structures. The expression of *AmqbHLH1* mRNA in *X. laevis* resulted in a dose-dependent reduction in or loss of ears and the lateral line system without eliminating ectopic neurons. At the lowest concentrations examined, we found that inner ear neurosensory development consisted sometimes of only a few scattered hair cells in a single-layer epithelium. Furthermore, low concentrations of *AmqbHLH1* mRNA affected inner ear afferent guidance. Our data suggest that the *AmqbHLH1* gene has some anti-neurosensory abilities in frogs and that the overexpression of a single gene may not be sufficient for stable long-term transdifferentiation in cells.

## 1. Introduction

Basic helix–loop–helix (bHLH) genes encode transcription factors that are important in defining cellular fate. In the central nervous system, bHLH genes are essential in neuron, oligodendrocyte, and astrocyte development [1,2]. Within peripheral neurosensory systems, bHLH genes are required for placodally derived neuron and hair cell development [3,4]. The *atonal* family of bHLH genes, which includes *neurogenin 1* (*Neurog1*), *neuronal differentiation 1* (*Neurod1*), and *atonal homolog 1* (*Atoh1*), plays an essential role in development across many different tissue types [5,6,7]. The origin of bHLH genes has been described principally through comparative data, and it has been suggested that the *atonal* family of bHLH genes originated early within the metazoan lineage [7,8]. In eumetazoan lineages, the *atonal* superfamily of bHLH genes underwent a series of duplication and diversification events, allowing each member of the family to evolve specific functions [9,10]. As a result, vertebrates utilize multiple *atonal* paralogs, mostly presented by *Neurog1*-like gene, that interact with *MycN* and E12/E47 (*Tcf3*) for DNA binding, transcription factor binding activity, E-box binding activity, and protein heterodimerization activity to generate distinct individual mechanosensory cells and sensory neurons [3,11,12].

Metazoan demosponges, such as *Amphimedon queenslandica*, have only a single *atonal*-like bHLH gene, *AmqbHLH1*, which is believed to be derived from the same proto-*atonal* gene as the eumetazoan *atonal* family paralogs, likely *Neurog1* [7]. However, sponges do not form any neurosensory or neuronal cell types beyond having a number of putative sensory cells [13,14]. Sponges, like choanoflagellate, have choanocytes that may evolve into hair cells [15,16,17,18]. Interestingly, the injection of sponge *AmqbHLH1* mRNA into the vertebrate *Xenopus laevis* induced ectopic neurons in the ectoderm, despite the absence of neuronal cell types in the sponge itself [19]. The formation of ectopic neurons following *AmqbHLH1* injection was reminiscent of neuronal transformation phenotypes observed following injection with *Neurog1*, *Neurod1*, or *Atoh1* mRNA [20,21,22]. Whereas the sponge itself lacks such cell types, the fact that *AmqbHLH1* can convert non-neuronal ectoderm in frogs into neuron-like cells seems to suggest an ability to override normal neuronal induction pathways. However, the extent to which this inductive influence maintains a stable and lasting transformation of the ectoderm into neurons beyond the early embryonic stages (*X. laevis* stages ~13–34) has not been explored. Furthermore, it is unknown whether such a transformation has developmental consequences for placodal-derived neurosensory structures, which are critically dependent on bHLH interactions within the ectoderm [4,23]. The inner ear, which only begins to develop around stage 21 [24] and the even later developing lateral line system of ectodermal neuromasts [25,26] rely on the correct timing and level of expression of several bHLH genes for the proper development of their neurosensory domain [27,28,29] within the specified ectoderm.

Given the specialized functions of the *atonal* superfamily genes in the neurosensory development of the inner ear and lateral line and the fact that *AmqbHLH1* expression induces the transformation of the ectoderm into neuron-like cells at periods that precede the development of the otic and lateral line placodes [26,30,31], we sought to study the effect of sponge *AmqbHLH1* on their development. The injection of sponge *AmqbHLH1* mRNA resulted in the formation of ectopic neuron-like cells in the embryo, as previously observed [19]. However, a stable transformation of these neurons was not observed at any of the doses, indicating that these ectopic neurons did not persist beyond their initial development. Nevertheless, despite there being no long-term neuronal transformations in the ectoderm, *AmqbHLH1* expression influenced the development of the later forming otic and lateral line placodes. Within the inner ear, reduced neurosensory development was most commonly observed following *AmqbHLH1* expression; however, it was observed at levels over an order of magnitude less than that of mouse *Neurod1* mRNA, which served as a reference for the effect of a less divergently related bHLH [21,32,33]. This suggests that the divergent bHLH sponge protein has strong anti-neurosensory activities within the cellular context of inner ear neurosensory development in *Xenopus*. To more comprehensively understand these developmental consequences of *AmqbHLH1* on the inner ear, particularly given the role of the closely related *Neurod1* atonal paralog in the central pathfinding of inner ear afferents [7,34,35,36], we assessed the central navigational properties of these afferents. The expression of this bHLH gene resulted in aberrant central projections of the remaining inner ear afferents, suggesting that all neurosensory components of the inner ear were affected by *AmqbHLH1* expression. Similarly, *AmqbHLH1* mRNA negatively affected lateral line development, though at higher doses than those required to disrupt the development of the inner ear.

## 2. Results

### 2.1. AmqbHLH1 Effects on the Long-Term Development of Ectopic Neurons in the Skin

Given the previously reported presence of neuronal markers in the ectoderm of neurula stage *X. laevis* following the injection of 500 pg of *AmqbHLH1* mRNA [7,19], we sought to determine whether the expression of these markers corresponded to the conversion and long-term sustainability of ectopic neurons in the skin of similarly injected tadpoles. The injection of 500 pg of *AmqbHLH1* mRNA resulted in tadpoles with severe phenological malformations (Appendix A). We next wanted to rule out the resultant phenotypes possibly being attributed to the disruption of endogenous translational machinery. Thus, lower doses were also examined using the injection of one (of two [24]) doses. Injections of 25 pg and 50 pg of *AmqbHLH1* mRNA were found to be able to induce ectopic neurons as had previously been demonstrated for 500 pg of *AmqbHLH1* mRNA [19]. Eight out of eleven animals unilaterally injected with 25 pg of *AmqbHLH1* mRNA and all ten animals unilaterally injected with 50 pg of *AmqbHLH1* mRNA expressed the neuronal marker *n-tubulin* on the injected side (Figure 1A’).

To determine whether these ectopic neurons were stable, *Xenopus* embryos unilaterally injected with either 25 pg, 50 pg, 125 pg, or 500 pg of *AmqbHLH1* were reared until the early tadpole stages (stage 46 [24]), where cranially located ectodermal neurons were fluorescently visualized with acetylated tubulin antibody. Ectopic ectodermal neurons were not observed on the injected side of these tadpoles (Figure 1B–E’). Indeed, there was no significant difference in the number of hair cells present in the skin between the control side and injected side (Figure 1B–E’) as indirectly detected by the intensity of tubulin staining (Mann–Whitney non-parametric, *p* > 0.05; three regions examined in each of the four animals). The fact that additional neurons were not found following the overexpression of *AmqbHLH1*, despite the overwhelming presence of neuronal markers in the ectoderm of neurula-stage embryos (Figure 1A,A’), suggests a lack of long-term sustainability in the ectopic neurons following developmental progression. These results suggest that a single, early overexpression of a bHLH gene is not sufficient for the sustained neuronal transformation of the ectoderm in *X. laevis*.

### 2.2. Effect on Lateral Line Neurosensory Development

The lateral line system of aquatic vertebrates, which shares developmental and cell type characteristics with the inner ear, also requires bHLH genes for neurosensory development [31,37]. We next sought to determine whether the injection of *AmqbHLH1* mRNA affected the development of the lateral line system. Embryos were injected with 25 pg, 50 pg, 125 pg, or 500 pg of AmqbHLH1mRNA. The development of the lateral line was investigated in stage 46 tadpoles using antibodies against acetylated tubulin to label nerve fibers and MyoVI to label hair cells. At the highest dose, 500 pg of *AmqbHLH1* mRNA, the supraorbital lateral line was severely truncated or even absent on the injected side (Figure 2A–A”). When the supraorbital lateral line was present, the number of neuromasts in animals injected with 500 pg *AmqbHLH1* mRNA (n = 5) was significantly less than in control (n = 6) animals (*p* < 0.05) (Figure 2A–A”,G). Similarly, there was a significant reduction in the number of neuromasts in animals injected with 125 pg of *AmqbHLH1* mRNA (n = 5) as compared to controls (*p* < 0.05); however, the overall development of the lateral line was less disrupted than at 500 pg (Figure 2B–B”,G). In contrast, there was no significant difference in neuromast number between animals injected with 25–50 pg of *AmqbHLH1* mRNA (n = 6) and controls (Figure 2C–C”,G). In these animals, the only observed abnormality was the presence of lateral line neuromasts in the space not occupied by the ear as determined by the approximation of anterior and posterior lateral lines (see white asterisks, Figure 2C,C”). However, the presence of lateral line neuromasts in this region was also observed following physical ear removal both in this study (Figure 2D–D”; n = 11) and as previously reported [38,39], suggesting that the presence of lateral line neuromasts in this region is not due to the expression of *AmqbHLH1* but rather due to the absence of, or a reduction in, the ear. Whereas there was an effect on the number of neuromasts in the lateral line (Figure 2G), *AmqbHLH1* expression does not affect the number of hair cells within a given neuromast. There was no significant difference in the number of hair cells within neuromasts in the injected side (Figure 2F) as compared with the uninjected side (Figure 2E) at any of the doses of *AmqbHLH1* mRNA (hair cells were counted in six neuromasts from each of the four animals).

### 2.3. Effect on Inner Ear Development

The injection of *AmqbHLH1* directed developmental deformations in major cranial sensory organs, even at the lowest doses (Figure 3A–A”’). Among the affected sensory organs was the inner ear, which itself is critically dependent upon bHLH genes for proper development [6]. We therefore wanted to determine whether this gene derived from the proto-atonal gene, *AmqbHLH1*, affected the formation of neurosensory cell types in the inner ear, particularly given its proneural capabilities, in particular *Neurod1* [33]. A total of 74 embryos injected with 25 pg of *AmqbHLH1* mRNA, 116 embryos injected with 50 pg of *AmqbHLH1* mRNA, 35 embryos injected with 125 pg of *AmqbHLH1* mRNA, and 12 embryos injected with 500 pg *AmqbHLH1* mRNA were examined at stage 46. All doses showed some degree of derailment of ear development (Table 1). Embryos that had defects in ear development were classified into three distinct phenotypes based on the degree of ear development following the injection of *AmqbHLH1* mRNA (Figure 3A–A”’):*No Ear.* Embryos had no recognizable inner ear development (Figure 3A,A”’).*Ear Vesicle.* Embryos had an ‘empty’ inner ear vesicle that was devoid of otoconia (Figure 3A’).*Reduced Ear.* Embryos had smaller inner ear development relative to the control side, but it contained otoconia (Figure 3A”).

The fewest cases showing the total loss of ear development occurred for the lowest dose administered, 25 pg *AmqbHLH1* mRNA, suggesting that the derailment of development was likely dose-dependent. Since there was such a profound effect on ear development with low concentrations of *AmqbHLH1* mRNA, mouse *Neurod1* mRNA was injected into *X. laevis* embryos to compare the effect of *AmqbHLH1* mRNA to the more closely related mouse bHLH gene, *Neurod1*. A total of 10 embryos injected with 25 pg mouse *Neurod1* mRNA, 14 embryos injected with 500 pg mouse *Neurod1* mRNA, 90 embryos injected with 1400 pg mouse *Neurod1* mRNA, and 93 embryos injected with 2800 pg of mouse *Neurod1* mRNA were analyzed at Stage 46 for ear phenotypes (Table 1). In contrast to *AmqbHLH1* mRNA, no noticeable ear-reduction phenotype was detected after an injection of 25 pg or even 500 pg (Figure 3B) of mouse *Neurod1* mRNA. Only when 1400 pg or 2800 pg of mouse *Neurod1* mRNA was injected was there an effect on ear development (Figure 3B’–B”’; Table 1). As with *AmqbHLH1*, a dose of mouse *Neurod1* mRNA had an effect with respect to the reduction in, or absence of, an ear, albeit at a concentration of mRNA over an order of magnitude higher. These data suggest that the overexpression of *AmqbHLH1* mRNA and mouse *Neurod1* mRNA negatively affects inner ear neurosensory development, and in a dose-dependent manner, but that the more distantly related *AmqbHLH1* mRNA is more potent than mouse *Neurod1* mRNA in affecting ear development in *X. laevis*.

We next examined the degree of neurosensory development in the ear, or in the otic region, following the injection of *AmqbHLH1* mRNA using antibodies against Tubulin to label nerve fibers and MyoVI to label hair cells. In embryos that had no inner ear development following the injection of either 25 pg or 50 pg of *AmqbHLH1* mRNA (n = 9), there were no MyoVI-positive cells on the injected side, confirming the complete absence of inner ear hair cells’ sensory epithelia. In embryos with no ear, there was an increased number of neurons of unknown origin and a massive disorganization of these neurons in the presumptive otic region (Figure 4A,B). This was in contrast to the examination of local otic regions for cranial nerve arrangement following the physical removal of the embryonic inner ear, where no such disorganization was observed (Figure 4C; [38]). Compared with the normal organization of sensory fibers after the removal of an ear (Figure 4C), the disorganized fibers did not follow a clear pattern of innervation (Figure 4A’,B). This suggests additional effects of AmqbHLH1 protein on neuronal development in the peripheral nerves, possibly mediated by altered neuronal migration and projection. In embryos injected with either 25 pg or 50 pg *AmqbHLH1* mRNA that had ear vesicles devoid of otoconia (n = 5), there were either no or very few hair cells (Figure 4D,E). Hair cells, if present, were randomly distributed and often found in a single-layer epithelium (Figure 4D,D’), an atypical characteristic of inner ear sensory epithelia (compare Figure 4A, left). Only occasionally was a canal crista observed (Table 2).

In tadpoles of the third phenotypic group, with ears of reduced size that contained otoconia following the injection of *AmqbHLH1* mRNA (n = 17), there were a range of degrees of development in the sensory epithelia (Table 2). In the most severely reduced phenotypes, ears contained a small patch of hair cells aggregated in the ventral portion of the ear or even entirely lacked hair cells. The highest frequency phenotype of reduced ears contained one to three semicircular canal cristae and a single gravistatic epithelium, formed by the lack of segregation of the utricle, saccule, and lagena (Figure 4F,G). In two of the less reduced ears, there were two distinct gravistatic epithelia where the utricle was segregated from the saccule–lagena. Overall, there were significantly fewer hair cells in embryos with reduced ears (n = 31.6 ± 6.7) compared with control ears (n = 89.7 ± 7.7) (*p* < 0.001, n = 17 animals, Mann–Whitney non-parametric test). Although the percentage reduction in the number of hair cells in reduced ears (62.3 ± 5.9%) was slightly larger than the percentage reduction in the total volume of the reduced ears (52.7 ± 15.3%), these percentages were not significantly different, indicating that hair cell loss happened at a similar rate relative to the total volume reduction. Moreover, MyoVI-positive cells were found occasionally in the ganglia (Figure 4G’). This suggests that *AmqbHLH1* may affect inner ear neurosensory development through interference with endogenous bHLH transcription factors, similarly to the known interaction of *Neurod1* with *Atoh1* [27,29].

### 2.4. Effect on Inner Ear Afferent Central Pathfinding

Given the disrupting influence on inner ear neurosensory development, and knowing that the conditional knockout of the bHLH gene *Neurod1* disrupts central pathfinding in surviving inner ear afferents in mice but knocking out *Atoh1* does not [29,35,40,41], we sought to determine whether the expression of *AmqbHLH1* would have an effect on inner ear sensory neuron central pathfinding in the hindbrain. Because we need an ear present to determine its central projections, we opted to inject only the lower 25 pg dose of *AmqbHLH1* mRNA as this dose resulted in more partially differentiated ears than higher concentrations (Figure 4). In frogs and other aquatic vertebrates, the vestibular nucleus is bordered at the dorsal boundary by the lateral line nuclei and at the ventral boundary by the trigeminal projections [31,42]. We therefore implanted lipophilic dyes into the inner ears, as well as the anterior lateral line and trigeminal nerves (Figure 5). In control animals, vestibular afferents projected exclusively between the lateral line afferent and trigeminal afferent projections (n = 12; Figure 5A). In contrast, vestibular afferents from animals injected with *AmqbHLH1* mRNA projected not only to the vestibular nuclei but also dorsally into the lateral line nuclei (n = 8; Figure 5B). In addition, some vestibular afferents could be observed in the most dorsal region of the trigeminal projection in these animals (Figure 5B).

Since the aberrant inner ear central projections observed in these *AmqbHLH1* mRNA-injected animals could be due to the effects of this transcription factor within the afferents themselves, within hindbrain nuclei, or both, we performed ear transplantations to differentiate between these possibilities. This method has been used previously in *X. laevis* to differentiate the central and peripheral mechanisms of afferent pathfinding characteristics [43]. Here, to segregate the hindbrain-specific effects of *AmqbHLH1* from the inner ear afferent effects of *AmqbHLH1*, an ear from an injected animal was swapped with one from a control and vice versa. This resulted in one group of animals in which everything, but the ear and its associated afferents was expressing *AmqbHLH1* mRNA (which assessed *AmqbHLH1* in the hindbrain alone) and a second group of animals in which only the ear and its associated afferents were expressing *AmqbHLH1* mRNA (which assessed *AmqbHLH1* in the inner ear alone). Inner ear afferents from control ears that were projecting into the hindbrain of an animal injected with *AmqbHLH1* mRNA projected directly and only to the vestibular nuclei (n = 8; Figure 5C), almost indistinguishable from wild-type control animals. In contrast, inner ear afferents from *AmqbHLH1* mRNA-injected ears transplanted into a control animal projected in a manner like that of animals injected with *AmqbHLH1* mRNA without ear transplantation. Inner ear afferents in these animals not only projected to the vestibular nucleus but also overlapped with lateral line and trigeminal projections (n = 9; Figure 5D). These data suggest that it is the expression of *AmqbHLH1* in the ear, likely along with the inner ear sensory neurons themselves, that affects central pathfinding.

## 3. Discussion

Our study expressing the atonal-like *AmqbHLH1* gene from the demosponge *Amphimedon queenslandica* in *X. laevis* expands upon previous work using the same approach that was previously limited to the neurulation stages [19], in line with similar studies assessing mammalian proneural bHLH transcription factors [20,21,22], by assessing development at later time points. While the expression of *AmqbHLH1* mRNA in *X. laevis* resulted in the expression of neuronal markers in the ectoderm in early stages [19] (Figure 1), we saw no evidence of excess neurons in the skin at later developmental stages (Figure 1). Our data therefore indicate that the initial transformation of the ectoderm into neuronal-like cells is a transient effect, similar to replacing *Atoh1* with *Neurog1* [44]. In part, this might relate to the fact that the initial dose of *AmqbHLH1* mRNA and subsequent translated protein would be degraded by endogenous mechanisms and thus be diluted over time. This suggests that the maintenance of these ectopic neurons might require the continued expression of *AmqbHLH1* mRNA, or other bHLH genes, above a certain threshold. Normally the expression of bHLH genes oscillates within and between cells, including DNA binding, transcription factor binding activity, E-box binding activity, and protein heterodimerization activity, and different levels of expression that correlate with the differentiation of specific cell types, such as specific *Atoh1* and *Neurog1* timing and levels for inner and outer cochlear hair cells [12] and the sponge [7]. Consistently with a progressive reduction in mRNA signal over time, effects on the neurosensory development of the inner ear and lateral line due to the expression of *AmqbHLH1* mRNA co-varied with the presumed levels of expression: while lateral line development was unaffected by low levels of *AmqbHLH1* expression, the ear was affected (Figure 2, Figure 3 and Figure 4). This may be due to the maturation of the ear and lateral line, including neuromast migration, occurring at different times [45,46]; with the lateral line system developing later, the ability of *AmqbHLH1* expression to alter neurosensory development likely depends upon the concentration available at the onset of placodal induction [4,47]. Nevertheless, our in vivo data highlight that there is no stable transformation of the ectoderm into neurons, implying that these cells can be forced to express neuronal markers but not transformed into a stable neuronal or hair cell phenotype without the cascade of gene expressions now known for placode or neuronal development [23,48].

In contrast to the transient induction of ectopic neurons, *AmqbHLH1* exhibited a potent anti-neurosensory effect in the inner ear and lateral line, seemingly in a dose-dependent manner (Table 1, Figure 2, Figure 3 and Figure 4). Similarly, the overexpression of *Neurod1* from mice resulted in an anti-neurosensory effect in the inner ear (Figure 3); however, the concentration needed to derail ear development with *Neurod1* mRNA was over an order of magnitude higher than that for *AmqbHLH1*. This suggests that *AmqbHLH1* is a more potent inhibitor of neurosensory development in the ear and is retained at high enough levels through these stages. The atonal family of bHLH transcription factors (*Atoh1*, *Neurod1*, and *Neurog1*) form complex protein interactions and oscillations in expression to determine whether cells undergo differentiation or proliferation [3,6]. Upstream of Neurog1, Neurod1, and Atoh1 expression is *Eya1/Six*, which also regulates with *HDACs*, *Rest*, and *Smarca4* to modulate chromatin and regulate neuronal development [49,50], the atonal family of bHLH transcription factors form heterodimers with E-proteins for appropriate DNA binding to specific E-boxes [51]. In addition, these E proteins can interact with *Hes/Hey* factors and inhibitors of DNA binding (*Id*) factors, which limit the ability of E proteins to heterodimerize to the bHLH transcription factors to regulate *Delta/Notch* [44,52]. It is possible that the addition of the *AmqbHLH1* transcription factor derails the existing bHLH transcription factor network in *Xenopus*, thus altering the normal balance between the native bHLH transcription factors and their interaction. For example, AmqbHLH1 transcription factors may siphon off additional E-proteins, leaving endogenous bHLH transcription factors disabled from signaling. Alternatively, to siphoning off E-proteins, *AmqbHLH1* transcription factors may heterodimerize with endogenous E-proteins and subsequently bind to specific E-box sites on the DNA, initiating a downstream signaling cascade or blocking others. Recently, it has been shown that the specific heterodimerization of bHLH transcription factors with E-proteins can alter the neurogenic strength of bHLH transcription factors based on the specific E-box that they bind to [53]. Given that evolutionary modification to E-box sequence elements has allowed differential binding and functional capabilities in endogenous bHLH transcription factors [51,54], it is possible that the available E-box sites in the *Xenopus* genome permit *AmqbHLH1* to bind preferentially among regulatory networks devoted to driving terminal differentiation. Regardless of the mechanism, the disruption of the proliferation of the otocyst or lateral line and early differentiation could result in the observed phenotype with reduced to no sensory tissue present. The specific mechanism remains to be explored.

The observed degree of sensory epithelia development within the inner ear was variable with respect to the correct segregation of specific endorgans (Table 2), with fusions and supernumerary epithelia patches. Likewise, there was variability in the number of neuromasts in the supraorbital lateral line, with fewer neuromasts present at the higher concentrations of *AmqbHLH1* mRNA. During development, and in an apparent recapitulation of evolution, the segregation and diversification of epithelia patches is a necessary precursor for the appropriate development of endorgans [34,55,56]. Improper segregation in reduced ears would imply yet another anti-neurosensory effect in the inner ear, known to result in mixed hair cell type development in certain mouse mutants [57,58]. The functional consequences of such a lack of segregation in frogs would be of interest but is beyond the scope of the present paper. Sparsely located hair cells in inner ear vesicles could be the result of the AmqbHLH1 transcription factor’s influence on regulatory networks associated with differentiation over proliferation, as mentioned above. We cannot exclude the possibility of AmqbHLH1 having an influence over the Notch/Delta pathway, which is known to critically influence hair cell formation [59,60]. This would be unsurprising, given the reported influence of AmqbHLH1 in setting up the Notch/Delta lateral inhibition process [19]. The fact that the number of hair cells within a given lateral line neuromast did not differ from the control might indicate either a different action of *AmqbHLH1* in these hair cells or alternatively that the concentration of *AmqbHLH1* mRNA was depleted as a function of time below the level that would affect neuromast hair cell differentiation at the time that these hair cells were differentiating. The presence of hair cells in the inner ear ganglia in some animals (Figure 4) might indicate a crosstalk between *AmqbHLH1* and the existing bHLH gene-regulatory network. A similar conversion of neurons into hair cells has previously been shown in mouse mutants of *Neurod1* [36], whereas *Atoh1*-expressing inner pillar cells [61] seem to be resistant to hair cell conversion [62] but can be converted for *Neurog1* misexpression [44]. These data further support the notion that the *atonal*-like *AmqbHLH1* gene is acting in a contextual manner, evident in variable actions in the ear and lateral line.

While the disorganization of the lateral line system in terms of the approximation of the anterior and posterior lateral lines can be explained purely by the absence of an ear (Figure 2 and Figure 3; [31]), the striking disorganization of the cranial nerves in animals lacking an ear following the expression of *AmqbHLH1* contrasts with the observed effects following physical ear removal. Physical ear removal had a minimal effect on the overall organization of the remaining cranial nerves (Figure 2C; [38]). While the principles that govern the neuronal disorganization following *AmqbHLH1* expression remain unclear, a possible conversion of otic placode tissue into neurons could support a limited proneural role [19]. Given that the lack of ear occurred more often with higher concentrations of *AmqbHLH1,* specifically those that were still an order of magnitude below the previous study on AmqbHLH1 [19], it remains possible that a pro- vs. anti-neuronal effect is both contextual and concentration-based.

Moreover, the expression of *AmqbHLH1* in *X. laevis* embryos resulted in derailed central pathfinding in existing inner ear afferents (Figure 5). The transplantation of ears confirmed that the expression of *AmqbHLH1* in the inner ear alone was sufficient for the derailment of afferents, suggesting that it is the expression within the neurons themselves that mediates improper targeting. Similarly, the conditional knockout of *Neurod1* in the inner ears or neurons of mice also resulted in aberrant central projections of inner ear afferents [35,36]. These data suggest that deviations from the normal level of bHLH gene expression negatively affect inner ear afferent central targeting. It is likely that these bHLH genes regulate the expression of downstream central guidance cues that provide spatial instruction for correct central termination along the dorsal–ventral axis [39,43,63,64]. Thus, it is compelling to suggest that AmqbHLH1 is either directly regulating the expression of downstream guidance molecules or altering the existing bHLH networks necessary for such regulation.

The forced reprogramming of cell types, as seminally described by Weintraub et al. [65], has previously raised hopes for inner ear cell-type transformations such as e.g., indiscriminate inner ear cells into lost neurons or hair cells [32,33,66]. However, such hopes have not yet proven biologically fruitful and lack translational success in generating hair cells out of supporting cells beyond a certain critical stage in postnatal mice [67,68,69,70,71]. Rather than forced transformation through the one-time overexpression of a single bHLH transcription factor [44,72], regulated and stable neuronal differentiation may depend on the proper timing of various expressions through feedback loops generating oscillations of expression [73]. For example, there is a differential time delay between cell cycle exit and the onset of *Atoh1* expression that correlates with hair cell viability [12], expanding on the emerging understanding of oscillation for normal development [54]. Our data strongly support the notion that the simple expression of a given bHLH transcription factor without taking all such variables into account may not lead to stable neuronal or hair cell differentiation but only to an apparently transient and unstable phenotype conversion with limited translational potential. Furthermore, not only may a single transcription factor not lead to the stable differentiation of a given cell type, it may also have detrimental effects on other systems critically dependent on the correct timing and expression level of that transcription factor or related factors.

## 4. Materials and Methods

### 4.1. Ethics Statement

All animal protocols used in these studies were approved by the Institutional Animal Care and Use Committee at the University of Iowa (#6031682). All experiments were performed in accordance with the relevant guidelines and regulations.

### 4.2. Animals

*Xenopus laevis* embryos were obtained through induced ovulation following an injection of human chorionic gonadotropin and fertilized with a sperm suspension in 0.3× Marc’s Modified Ringer’s Solution (MMR) and washed in 0.1× MMR (diluted from 1× MMR. Then, 1× MMR pH 7.6–7.8 was diluted from 10× stock (1 M NaCl, 18 mM KCl, 20 mM CaCl_2_, 10 mM MgCl_2_, 150 mM Hepes). All embryos were kept at 18 °C in 90 mm Petri dishes containing 0.1× MMR until they reached later neurula stages or stage 46 [24]. Embryos fixed at neurula stages were fixed in 4% formaldehyde in 1.1× MEM salts and washed in MeOH. Embryos fixed at stage 46 were anesthetized in 0.02% Benzocaine [74] and fixed in 4% paraformaldehyde (PFA) by immersion.

### 4.3. AmqbHLH1 and Mouse Neurod1 mRNA Injections

For the synthesis of sponge (*Amphimedon queenslandica*) *AmqbHLH1* mRNA [19] and mouse *Neurod1* mRNA, plasmid templates (pCS2+/AmqbHLH-1) were linearized using *Sac*II and purified and mRNAs were synthesized using SP6 RNA polymerase from the mMessage mMachine^TM^ transcription kit (Ambion; Thermo Fisher Scientific, Austin, TX, USA). The protocol was followed according to the manufacturer’s directions. Embryos used for the injection of *AmqbHLH1* or *Neurod1* were dejellied in 2% cysteine in 0.1× MMR pH 7.8 shortly after fertilization. Embryos were placed in a Ficoll solution (2% Ficoll 400, GE/Pharmacia, in 0.5× MMR) 5 min prior to injection and kept in the Ficoll solution overnight. *AmqbHLH1* or mouse *Neurod1* mRNA were diluted with RNase-free water so that the final amount of mRNA injected was 25 pg, 50 pg, 125 pg, or 500 pg for *AmqbHLH1* or 25 pg, 500 pg, 1400 pg, or 2800 pg for mouse *Neurod1*. Embryos were injected into one cell at the 2-cell stage using a calibrated glass needle controlled by a Pico-Injector (Harvard Apparatus, Holliston, MA, USA), resulting in the treatment of one half of the embryo and allowing the other half to serve as an internal control. Animals were classified according to the degree of ear development at stage 46 (Table 1). For ear transplantations (see below), embryos were injected into both cells at the 2-cell stage with 25 pg *AmqbHLH* mRNA.

### 4.4. In Situ Hybridization

Following fixation, late-neurula-stage embryos (stage 20–21; Figure 1) injected with 25 pg (n = 11) or 50 pg (n = 10) *AmqbHLH1* mRNA, as well as uninjected controls (n = 12), were used for in situ hybridization for the neuronal marker *n-tubulin/tubb2b* [19,75]. Whole mount in situ hybridization was performed as previously reported [76]. The RNA probe was synthesized as described [77] using supplied enzymes and buffers from Promega. Digoxigenin-labeled RNA probe against n-tubulin was used at 1 µg/mL. Anti-digoxigenin-AP antibody (Roche Applied Science, Roche, Germany) was used at 1:2500. The hybridization of the probe was detected with BM Purple (Roche Applied Science), stopped with Bouin’s fixative, washed multiple times in 70% EtOH/10 mM Tris, pH 8.0, and bleached as described [77].

### 4.5. Ear Transplantations

Transplantations were performed in 1× MMR. Single otic placodes from a stage 25–27 [24] *X. laevis* injected with *AmqbHLH1* mRNA were removed using fine tungsten needles and transplanted into age-matched, uninjected control hosts, replacing the native ear of the same side (n = 25), as previously described [43]. In addition, single otic placodes were transplanted from uninjected control animals to replace the native ear of animals injected with *AmqbHLH1* mRNA at stage 25–27 (n = 24). Ear removals for lateral line analysis were also performed at stage 25–27 (n = 2) as previously described [38,78]. Wounds were allowed to heal for approximately 30 min in 1× MMR before animals were transferred to 0.1× MMR for subsequent rearing.

### 4.6. Immunohistochemistry

Following fixation in 4% PFA at least overnight, stage 46 embryos were dissected to remove the lower jaw, brains, and skin, leaving just the dorsal skin attached at the rostral end of the head, washed in PBS for three washes of ten minutes each, and then blocked in 5% NGS/0.1% Triton X-100 (Sigma-Aldrich, Munich, Germany) for one hour. Primary antibodies were used against acetylated tubulin to label all nerves [79] and myosin VI (MyoVI) to label hair cells [80]. The concentration used for acetylated Tubulin (Sigma-Aldrich, Munich, Germany) was 1:800, and that used for MyoVI (Proteus BioSciences, Waltham, MA, USA) was 1:400. Primary antibodies were diluted in PBS. Embryos were incubated in primary antibody for 72 h at 4 °C or overnight at 36 °C. Embryos were washed three times for ten minutes each and again blocked in the above blocking buffer for an hour. Species-specific secondary antibodies conjugated to Alexa dyes were used at 1:500 and were diluted in PBS. Embryos were incubated in secondary antibody and Hoechst nuclear stain overnight at 4 °C. Embryo heads were washed six times for fifteen minutes each and were mounted separately from their dorsal skin on a slide in glycerol and imaged with a Leica TCS SP5 or SP8 (Leica System, Wetzlar, Germany) confocal microscope with LAS-X software, version 3.5.

### 4.7. Quantification of Ectodermal Neurons

The dorsal skin of the head was mounted in glycerol on a microscope slide. Confocal z-series images at 2 µm were taken of the skin using a Leica TCS SP8 confocal microscope. The stacks of z-series images were collapsed and exported as TIFF files, and three boxes (100 µm × 100 µm) were selected at random from the non-lateral-line-containing ectoderm per animal (n = 4 animals per mRNA dose). New TIFF images were saved from cropping along the boxes. The average fluorescent intensity was calculated as previously described using the histogram function in ImageJ software, version1.5 (Teaching Image Processing, free available [43]).

### 4.8. Lipophilic Dye Tracing

For vestibular afferent labeling, the lower jaw of fixed stage 46 embryos was removed and pieces of lipophilic dye-soaked filter paper were flattened and implanted ventrally into both ears (NeuroVue^®^ Maroon, Exton, PA, USA), as well as into the trigeminal/anterior lateral line ganglia (NeuroVue^®^ Red, Molecular Targeting Technologies, Inc., West Chester, PA, USA) as previously described [39,43,81]. Animals were placed in vials of 4% PFA at 36 °C overnight to allow for dye diffusion. In total, 12 control animals, 8 animals injected with *AmqbHLH1* mRNA, 8 animals with a control ear transplanted into an animal injected with *AmqbHLH1* mRNA, and 9 animals with an ear from an animal injected with *AmqbHLH1* mRNA transplanted into a control animal were analyzed. Hemisected brains were flat-mounted on a slide in glycerol. Images were acquired using a Leica TCS SP5 confocal microscope with LAS-X software version 3.5.

### 4.9. Statistical Tests

The results were analyzed using the Mann–Whitney and Kruskal–Wallis non-parametric tests. Significance was determined at *p* < 0.05.

### 4.10. Data Availability

The data that support the findings of this study are available from the corresponding author upon reasonable request.

## 5. Conclusions

bHLH genes are ancient and found in sponges and vertebrates that multiply from about 20 to over 100 genes [7,17]. Among the partially conserved bHLH genes are *atonal*-like, *MycN*, and *Tcf3*, which allow DNA-binding transcription factor binding activity [49,50], E-box binding activity, and protein heterodimerization activity [54]. Here, we show that the injection of sponge gene *AmqbHLH1* into developing frogs causes a transient phenotype conversion of ears and neuromasts that will lose these sensory organs entirely with higher concentrations. The next steps will involve exploring the delayed expression of *AmqbHLH1* to fully understand gene interactions with native frog genes that also have similarities with mouse *Neurod1* in early expression.

## Figures and Tables

**Figure 1 ijms-26-05487-f001:**
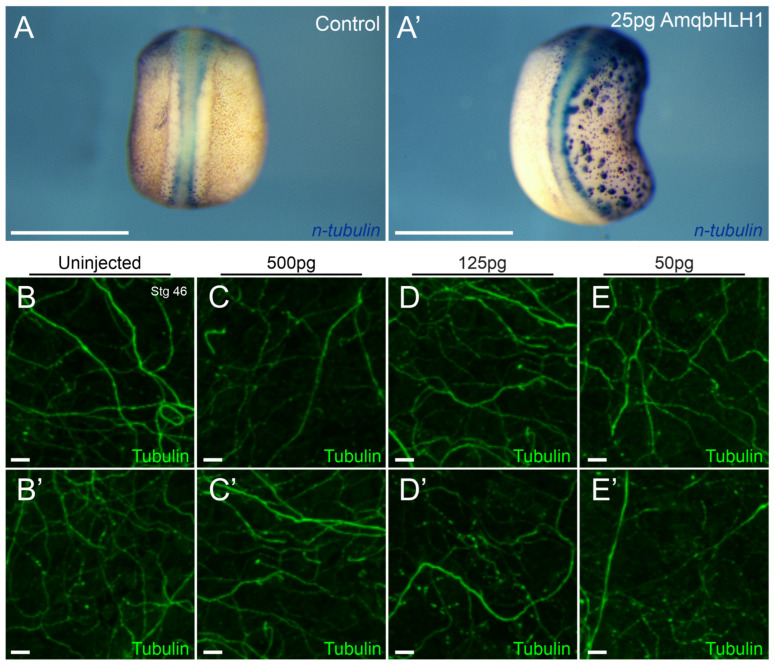
*AmqbHLH1* mRNA does not have a long-term ectopic neuron effect in the ectoderm. (**A**,**A’**) In situ hybridization for *n-tubulin* expression in control neurula-stage embryos (**A**) and in embryos injected on the right side with 25 pg *AmqbHLH1* mRNA (**A’**) (eight of eleven animals showed positive staining at 25 pg of stage 20); (**B**–**E’**) 100 µm × 100 µm areas of representative cranial ectoderm of stage 46 tadpoles stained with acetylated tubulin antibody. Images show non-lateral line neurons present in the skin in uninjected halves (**B**,**B’**) and in halves injected with 500 pg *AmqbHLH1* mRNA (**C**,**C’**), 125 pg *AmqbHLH1* mRNA (**D**,**D’**), and 50 pg *AmqbHLH1* mRNA (**E**,**E’**). Scale bar is 1 mm in (**A**,**A’**) and 10 μm in (**B**–**E’**).

**Figure 2 ijms-26-05487-f002:**
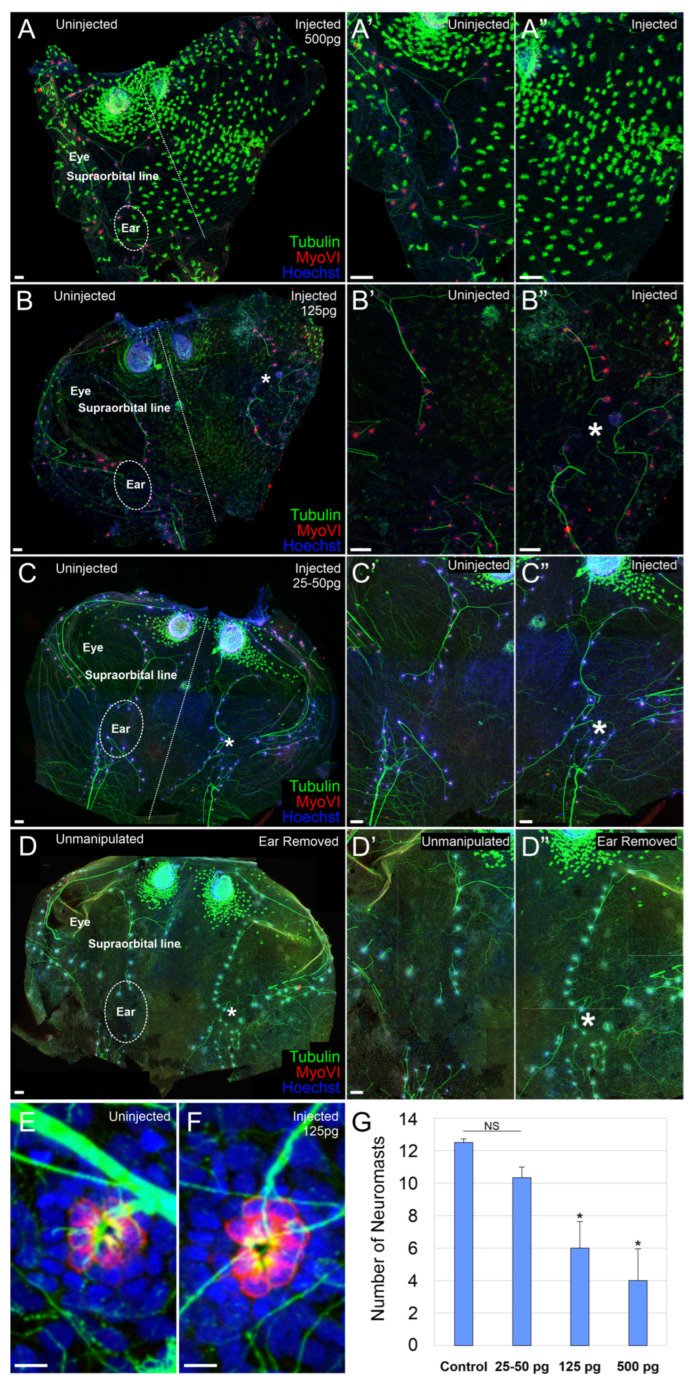
*AmqbHLH1* mRNA affects the lateral line. (**A**–**C**) Immunohistochemistry using antibodies against acetylated tubulin (green) and MyoVI (red) from an animal in which the right half was injected with 500 pg (**A**), 125 pg (**B**), or 25–50 pg (**C**) *AmqbHLH1* mRNA. (**A’**,**A”**) Higher magnification of the uninjected and injected halves, respectively, of (**A**), showing the absence of a supraorbital lateral line on the injected side. (**B’**,**B”**). Higher magnification of the uninjected and injected halves, respectively, of (**B**), showing some abnormal development in the lateral line. (**C’**,**C”**). Higher magnification of the uninjected and injected halves, respectively, of (**C**), showing near-normal development in the lateral line. (**D**) Immunohistochemistry using antibodies against acetylated tubulin (green) and myoVI (red) from an animal in which the right ear was physically removed at stage 25–27. (**D’**,**D”**) Higher magnification of the unmanipulated and ear-removed halves, respectively, of (**D**). For both types of manipulations in (**C**,**D**), there are no lateral line neuromasts over the ear (circled); however, when the ear was absent/reduced following *AmqbHLH1* mRNA injection or absent following physical ear removal, there was an expansion of lateral line neuromasts into the region previously occupied by the ear (white asterisks). (**E**) Single neuromast from an uninjected side. (**F**) Single neuromast from an animal injected with 125 pg *AmqbHLH1* mRNA. (**G**) Number of neuromasts in the supraorbital line of the lateral line system, ± SEM, * *p* < 0.05; NS, not significant. Scale bars are 100 μm in (**A**–**D”**) and 10 μm in (**E**,**F**). Skins were counterstained with the nucleus marker Hoechst, dash line in (**B**,**C**) show the boundary of left and right side. (**E**) at any of the doses of *AmqbHLH1* mRNA (hair cells were counted in six neuromasts from each of the four animals).

**Figure 3 ijms-26-05487-f003:**
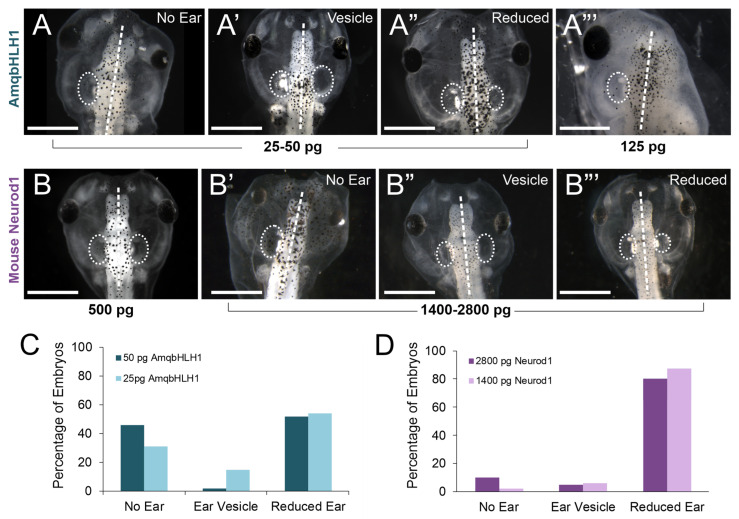
Phenotype assessment of ear development following *AmqbHLH1* or mouse *Neurod1* mRNA expression. (**A**–**A”**) Stage 46 *Xenopus laevis* tadpoles showing the no-ear, ear-vesicle, and reduced-ear phenotypes, respectively, following the injection of either 25 pg or 50 pg of *AmqbHLH1* mRNA into the right half of the animal. (**A’”**) Stage 46 *X. laevis* tadpole showing no ear, as well as no eye, on the *AmqbHLH1* mRNA-injected (right) side. (**B**) Stage 46 *X. laevis* tadpole showing no effect following injection of mouse *Neurod1* mRNA. (**B’**–**B’”**) Stage 46 *X. laevis* tadpoles showing the no-ear, ear-vesicle, and reduced-ear phenotypes, respectively, following the injection of either 1400 pg or 2800 pg of mouse *Neurod1* mRNA into the right half of the animal. Scale bars are 1 mm. (**C**) Percentage of embryos with the no-ear, ear-vesicle, or reduced-ear phenotypes for 25 pg and 50 pg of *AmqbHLH1* mRNA. (**D**) Percentage of embryos with the no-ear, ear-vesicle, or reduced-ear phenotypes for 1400 pg and 2800 pg of mouse *Neurod1* mRNA. Dot circles indicate the size of ears, dash line indicate the left/right side. Scale bars in (**A**–**B’’’**) are 1 mm.

**Figure 4 ijms-26-05487-f004:**
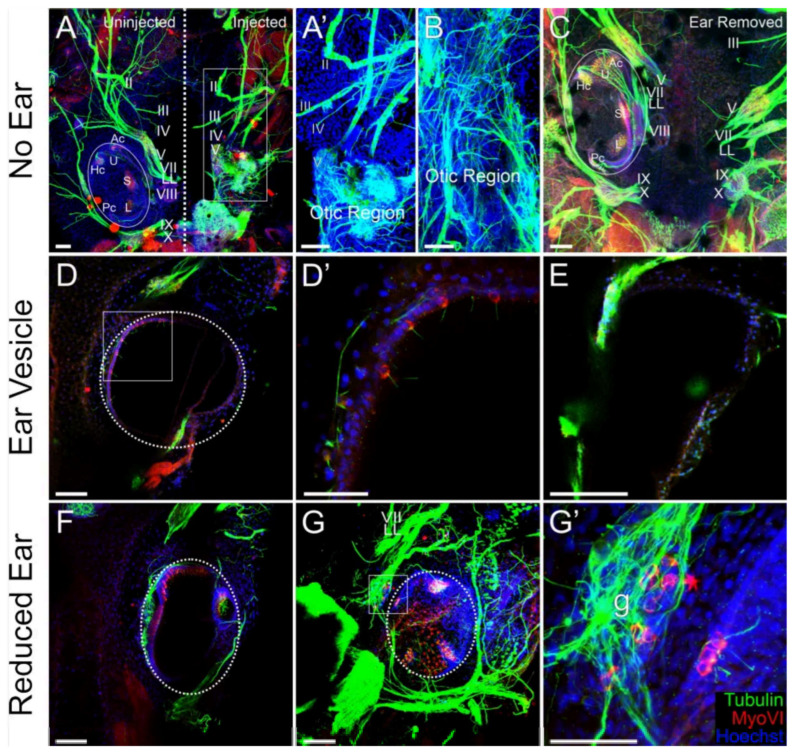
Low doses of *AmqbHLH1* mRNA disrupt the neurosensory development of the inner ear. (**A**–**G’**) Immunohistochemistry using antibodies against acetylated tubulin (green) and MyoVI (red) in stage 46 tadpoles unilaterally injected with 25 pg or 50 pg of *AmqbHLH1* mRNA. (**A**,**B**) Immunohistochemistry confirming the complete loss of the inner ear and instead showing a mass of neurons in the otic region on the injected (right) side in two animals in which no ear was detected. The anterior is at the top in all images. (**A’**) Higher magnification of the boxed area in (**A**). (**C**) An animal in which an ear was physically removed reveals an approximation of pre- and post-otic cranial ganglia but no overall disorganization. (**D**,**E**) Immunohistochemistry revealed limited (**D**) to no (**E**) neurosensory development in two animals with an ear vesicle on the injected side. (**D’**) Higher magnification of the boxed area in (**D**), showing a few scattered hair cells in a single-layer epithelium. (**F**,**G**) Immunohistochemistry showed a lack of segregation of the utricle–saccule–lagena in two animals with reduced ears. (**G’**) Higher magnification of the boxed area in (**G**), showing the presence of myoVI-positive cells in the otic ganglia (g). Abbreviations: Ac, anterior canal; Hc, horizontal canal; L, lagena; Pc, posterior canal; S, saccule; U, utricle; II, optic nerve; III, oculomotor nerve; IV, trochlear nerve; V, trigeminal nerve; VII, facial; LL, lateral line; VIII, vestibular nerve; IX, glossopharyngeal nerve; X, vagus nerve. Dot circles indicate the size of ears, dash line indicate the left/right side, box is larger image (**D**,**D’**,**G**,**G’**). Scale bars are 100 μm in (**A**,**A’**,**B**–**D**,**E**–**G**) and 50 μm in (**D’**,**G’**).

**Figure 5 ijms-26-05487-f005:**
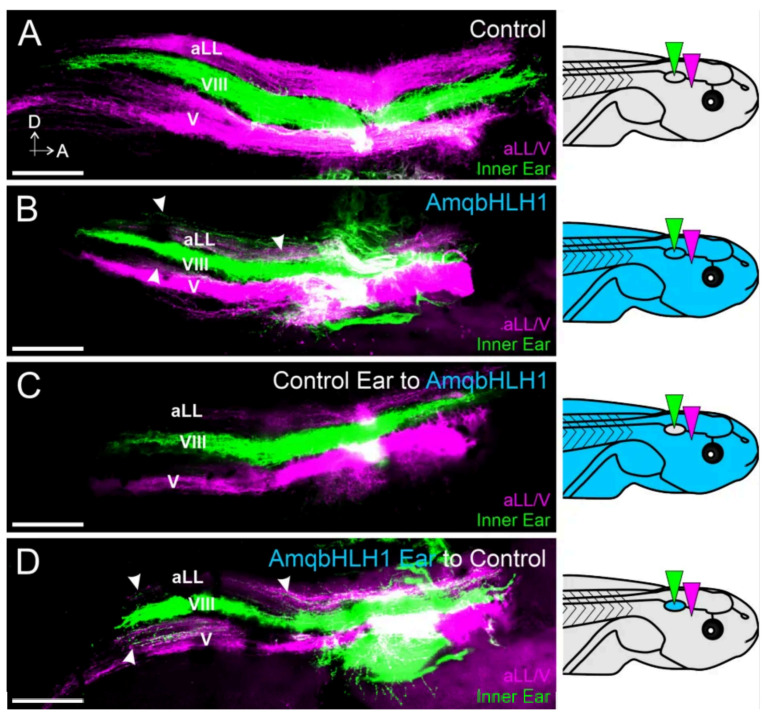
*AmqbHLH1* mRNA disrupts the central pathfinding of inner ear afferents. (**A**) Lipophilic dyes implanted into the ear (green) and into the trigeminal and anterior lateral line nerves (magenta) of a control animal demonstrate normal central projections. Grey and light blue background indicates tadpoles with or without mRNA. The inner ear afferent projections to the vestibular nucleus (VIII) are bordered dorsally by anterior lateral line afferents (aLLs) and ventrally by trigeminal afferents (V). The dorsal area is at the top and the anterior is on the right in all images. (**B**) Lipophilic dye implantation in the ear of an animal injected with *AmqbHLH1* mRNA reveals inner ear projections that not only project to the vestibular nucleus but also to the lateral line and trigeminal nuclei (arrowheads). (**C**) Lipophilic dye implantation in the ear from a control animal transplanted into an animal injected with *AmqbHLH1* mRNA reveals normal central projections. (**D**) Lipophilic dye implantation in the ear of an animal injected with *AmqbHLH1* mRNA transplanted into a control animal reveals inner ear projections that not only project to the vestibular nucleus but also to the lateral line and trigeminal nuclei (arrowheads). Scale bars are 100 μm. Diagrams represent treatment (gray, control; blue, *AmqbHLH* mRNA-injected), and colored wedges represent lipophilic dye placement (green, inner ear; magenta, anterior lateral line and trigeminal nerves).

**Table 1 ijms-26-05487-t001:** Scored ear phenotypes following injection of *AmqbHLH1* or *Neurod1* mRNA.

Dose of mRNA	Total	No Ear	Ear Vesicle	Reduced Ear	No Effect
25 pg *AmqbHLH1*	74	23	11	40	0
50 pg *AmqbHLH1*	116	53	3	60	0
125 pg *AmqbHLH1*	35	34	1	0	0
500 pg *AmqbHLH1*	12	12	0	0	0
25 pg *Neurod1*	10	0	0	0	10
500 pg *Neurod1*	14	0	0	0	14
1400 pg *Neurod1*	90	2	2	37	49
2800 pg *Neurod1*	93	9	3	52	29

Numbers in columns represent the numbers of animals per condition. Ear development was assessed as follows: (1) No Ear: Embryos had no recognizable inner ear development. (2) Ear Vesicle: Embryos had an ‘empty’ inner ear vesicle that was devoid of otoconia. (3) Reduced Ear: Embryos had less inner ear development relative to the control side but contained otoconia. (4) No Effect: No noticeable ear phenotype was observed.

**Table 2 ijms-26-05487-t002:** Degree of sensory epithelia development in animals with reduced ears following injection of *AmqbHLH1* mRNA.

Phenotype	U, Fused S-L, 3 Canals	Fused U-S-L, 4 Canals	Fused U-S-L, 3 Canals	Fused U-S-L 2 Canals	Fused U-S-L 1 Canal	Fused U-S-L No Canals	No HCs
Ear Vesicle	0	0	0	1	1	3	0
Reduced Ear	2	1	2	1	3	6	2

Numbers represent numbers of animals per condition. HCs, hair cells; L, lagena; S, saccule; U, utricle.

## Data Availability

The original contributions presented in this study are included in the article. Further inquiries can be directed to the corresponding authors.

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
