# Peer review of "Sponge bHLH Gene Expression in *Xenopus laevis* Disrupts Inner Ear and Lateral Line Neurosensory Development and Otic Afferent Pathfinding"

_ijms, 2025, doi:10.3390/ijms26125487_

Round 1
Reviewer 1 Report
Comments and Suggestions for Authors
Using xenopus embryos, this study found that the expression of sponge AmqbHLH1 gene is not sufficient for neuron maintenance and this gene has anti-neurosensory properties and influences afferent neuron pathfinding, providing insights into the role of sponge bHLH transcription factors in the development of neuron and sensory organ.
Major Points:
1. Comparative Analysis of AmqbHLH1 and NeuroD. Please provide detailed sequence information for both AmqbHLH1 and mouse NeuroD mRNA constructs used in the study and include molar concentration comparisons between the two mRNAs used in the experiments.
2. mRNA Stability and Detection. It is recommended to perform qPCR analysis to verify AmqbHLH1 mRNA persistence at later developmental stages.
3. Experimental Controls. Please incorporate injection of a non-functional mutant form of AmqbHLH1 as a negative control. This is crucial to distinguish specific gene effects from potential toxicity of mRNA injection.
Minor Points:
1. Figure 2 requires higher resolution images of the otic vesicle. Consider adding insets or enlarged views to better demonstrate key morphological features
2. The citation in Line 72 needs to be formatted consistently with the rest of the manuscript
3. Please verify all citations follow the journal's style guide. The format of journal name in references 5, 33, 35, 38, 43, and 50 need to be checked.
4. The first column in Table 1 need to be checked to correspond to the following content.
Author Response
Using xenopus embryos, this study found that the expression of sponge AmqbHLH1 gene is not sufficient for neuron maintenance and this gene has anti-neurosensory properties and influences afferent neuron pathfinding, providing insights into the role of sponge bHLH transcription factors in the development of neuron and sensory organ.
Thank you for the review, it is appreciated.
Major Points:
- Comparative Analysis of AmqbHLH1 and NeuroD. Please provide detailed sequence information for both AmqbHLH1 and mouse NeuroD mRNA constructs used in the study and include molar concentration comparisons between the two mRNAs used in the experiments.
No sequence was made but we have the dilutions of mRNA injected was 25pg, 50pg, 125pg, and 500pg for AmqbHLH1 or 25pg, 500pg, 1400pg, and 2800pg for mouse Neurod1
- mRNA Stability and Detection. It is recommended to perform qPCR analysis to verify AmqbHLH1 mRNA persistence at later developmental stages.
Sorry, the mRNA disappears with time and we cannot show the later expression at St. 46 old tadpoles.
- Experimental Controls. Please incorporate injection of a non-functional mutant form of AmqbHLH1 as a negative control. This is crucial to distinguish specific gene effects from potential toxicity of mRNA injection.
We do not inject with controls X. laevis that serves as the control (see Fig. 1). We show that concentration is increased with concentration. There is a potential effect from higher concentrations with toxicity, but we assume the highest concentration should be a specific deletion of neuronal development of the ear. Note that the different concentration requires a much higher injection for Neurod1 that rules out a toxicity effect with increased concentration.
Minor Points:
- Figure 2 requires higher resolution images of the otic vesicle. Consider adding insets or enlarged views to better demonstrate key morphological features
Please find attached the revised version that are enlarged to see the details. Thak you.
- The citation in Line 72 needs to be formatted consistently with the rest of the manuscript
Thank you we reformatted the citations.
- Please verify all citations follow the journal's style guide. The format of journal name in references 5, 33, 35, 38, 43, and 50 need to be checked.
- We double checked the citations, thank you.
- The first column in Table 1 needs to be checked to correspond to the following content.
Indeed, we double checked the Table and the corresponding Figures. Thank you for pointing up these details.
Reviewer 2 Report
Comments and Suggestions for Authors
In this manuscript, the authors studied the effects of AmqbHLH1 expression on inner and lateral line neurosensory development and found that it decreased the neurosensory ability in a dose-dependent manner. It provides insights into the study of ectodermal ectopic neurons in Xenopus Laevis. I have a few questions:
- On line 108, are you going to show the results of Fig. 4B-E’? If that’s the case, the organization of figures should be revised.
- You also studied the effect of mouse Neurod1 expression in X. Laevis, to compare to AmgbHLH1 expression, what effects does mouse Neurod1 expression have on mouse embryos? Is there a pattern or relationship?
- The figure 3 is unclear in explaining the context. Could you please show the single channel images as well to clarify the results?
- The references format: Please make the reference citation format consistent. e.g. do not capitalize each first letter in the title; use full name or abbreviation name of journal.
Author Response
In this manuscript, the authors studied the effects of AmqbHLH1 expression on inner and lateral line neurosensory development and found that it decreased the neurosensory ability in a dose-dependent manner. It provides insights into the study of ectodermal ectopic neurons in Xenopus Laevis. I have a few questions:
- On line 108, are you going to show the results of Fig. 4B-E’? If that’s the case, the organization of figures should be revised.
Please find attached the revised Fig. 5 in larger images.
- You also studied the effect of mouse Neurod1 expression in X. Laevis, to compare to AmgbHLH1 expression, what effects does mouse Neurod1 expression have on mouse embryos? Is there a pattern or relationship?
According to the latest paper we can assume that bHLH gene is more likely like Neurog1 not Atoh1. Moreover, Neurog1 is downstream of Neurod1, the injection used from mouse (See Ma et al., 1998). Moreover, we presented the newest review that details the sponge and mouse bHLH genes (Fortunato et al., 2016; Fritzsch et al., 2020).
- The figure 3 is unclear in explaining the context. Could you please show the single channel images as well to clarify the results?
Yes, we present the image as a larger image with single color for tubulin alone.
- The references format: Please make the reference citation format consistent. e.g. do not capitalize each first letter in the title; use full name or abbreviation name of journal.
Yes, we corrected the citations following the MDPI references.